# Genetic Neuropathy Due to Impairments in Mitochondrial Dynamics

**DOI:** 10.3390/biology10040268

**Published:** 2021-03-26

**Authors:** Govinda Sharma, Gerald Pfeffer, Timothy E. Shutt

**Affiliations:** 1Departments of Medical Genetics and Biochemistry & Molecular Biology, Cumming School of Medicine, Alberta Children’s Hospital Research Institute, Hotchkiss Brain Institute, University of Calgary, Calgary, AB T2N 4N1, Canada; govinda.sharma@ucalgary.ca; 2Departments of Clinical Neurosciences and Medical Genetics, Cumming School of Medicine, Hotchkiss Brain Institute, Alberta Child Health Research Institute, University of Calgary, Calgary, AB T2N 4N1, Canada; gerald.pfeffer@ucalgary.ca

**Keywords:** neuropathy, mitochondria, mitochondrial dynamics, fusion, fission, transport, quality control

## Abstract

**Simple Summary:**

Mitochondria are organelles within our cells that are best known for their role in energy production. They are also able to fuse, divide, and move within the cell (referred to as mitochondrial dynamics). This is especially important in neurons, where cells can be very long as they travel in peripheral nerves to send signals to muscles or detect sensory stimuli. Problems with mitochondrial dynamics can result in a spectrum of human diseases, ranging from milder disease in late adulthood through to severe, lethal, early onset diseases. In this review, we discuss the important genes involved in mitochondrial dynamics that have also been connected with genetic neuropathies. We explain how these gene products interact to maintain normal mitochondrial functions and describe some common themes, such as mitochondrial quality control.

**Abstract:**

Mitochondria are dynamic organelles capable of fusing, dividing, and moving about the cell. These properties are especially important in neurons, which in addition to high energy demand, have unique morphological properties with long axons. Notably, mitochondrial dysfunction causes a variety of neurological disorders including peripheral neuropathy, which is linked to impaired mitochondrial dynamics. Nonetheless, exactly why peripheral neurons are especially sensitive to impaired mitochondrial dynamics remains somewhat enigmatic. Although the prevailing view is that longer peripheral nerves are more sensitive to the loss of mitochondrial motility, this explanation is insufficient. Here, we review pathogenic variants in proteins mediating mitochondrial fusion, fission and transport that cause peripheral neuropathy. In addition to highlighting other dynamic processes that are impacted in peripheral neuropathies, we focus on impaired mitochondrial quality control as a potential unifying theme for why mitochondrial dysfunction and impairments in mitochondrial dynamics in particular cause peripheral neuropathy.

## 1. Introduction

### 1.1. Mitochondrial Dynamics

Mitochondria are highly active, double membraned organelles with many critical cellular functions. The term mitochondrial dynamics typically refers to the reciprocal processes of mitochondrial fusion and fission, which determine the morphology of the mitochondrial network. However, the term mitochondrial dynamics can also be expanded to include processes such as the transport of mitochondria [1], mitochondrial interactions with other organelles [2], as well as the biogenesis of new mitochondria, and the turnover of old or damaged mitochondria via mitochondrial autophagy (mitophagy) [3] or small mitochondrial derived vesicles [4]. Many of these dynamic processes are interrelated and depend on one another. For example, fission is required to generate fragments that are small enough to be degraded by mitophagy, or to move about the cell prior to fusing with other mitochondria. These dynamic processes allow mitochondria to adapt to physiological cues and are critical in maintaining cellular energetics, Ca^2+^ signaling, lipid biogenesis, mitochondrial quality control, and cell survival. 

### 1.2. Mitochondrial Dynamics in Neurological Disease

The importance of mitochondrial dynamics is highlighted by the fact that mouse knockout models of essential genes that are required for fusion (MFN1, MFN2 and OPA1) or fission (DNM1L) result in mitochondrial dysfunction and embryonic lethality [5,6,7,8,9]. As such, it should come as no surprise that impairments to mitochondrial dynamics are increasingly recognized to play a role in disease [10], in particular those affecting the brain, and that targeting these processes is gaining traction as a promising therapeutic approach [11]. 

While mitochondrial dynamics are important for all cell types, they are of critical importance in neuronal cells, which have high energy requirements. In this regard, continuous fusion and fission ensures a homogenous distribution of mitochondrial contents and is necessary for efficient oxidative phosphorylation. Meanwhile, the unique cellular morphology of neurons, which comprise long axons, also contributes to their reliance on mitochondrial dynamics. Notably, while neuronal mitochondrial biogenesis and degradation were once believed to occur only in the cell body or soma of the neuron, there is emerging evidence for biogenesis and degradation occurring within axons [12]. Nonetheless, mitochondria still need to be transported to and from the sites of high ATP demand such as synapses, which are mostly located in distal regions of axons, often at extraordinary distances from the cell body. Thus, efficient intracellular transport and quality control mechanisms, in parallel with mitochondrial fusion and fission, are critical for neuronal cell function.

Supporting their importance in the brain, impaired mitochondrial dynamics have been associated with multiple neurodegenerative diseases such as Alzheimer disease, Huntington disease, Parkinson disease, etc. [13,14,15]. Parkinson disease, in particular, is linked to impairments in mitophagy [16]. Meanwhile, specific pathogenic variants in a number of essential fusion (OPA1 and MFN2) and fission (DRP1) proteins lead to various neuronal pathologies in humans, such as optic atrophy, peripheral neuropathy, and encephalopathy [17,18,19,20]. Additionally, a growing list of pathogenic variants have also been identified in other proteins that are also involved in mediating mitochondrial fusion (FBXL4, MSTO1) [21,22,23] or fission (MFF, MID49) [24,25]. In this review, we will focus on genetic neuropathy as a common phenotype often associated with impaired mitochondrial dynamics (Table 1). 

### 1.3. Genetic Peripheral Neuropathy

Neuropathy occurs as part of the phenotype in numerous genetic diseases, either as a primary feature of the disease, or as part of a multisystem disorder. When neuropathy is the major feature of the clinical phenotype, the pattern of involvement allows categorization into one or more categories. When motor and sensory nerves are involved, this is referred to as hereditary motor and sensory neuropathy (HMSN), which is often used synonymously with Charcot Marie Tooth disease (CMT) [26]. Predominant motor involvement is referred to as hereditary motor neuropathy, or distal hereditary motor neuropathy (dHMN), which has substantial overlap with HMSN in its genetic causes [27]. Pure sensory involvement occurs in certain rare disorders referred to as hereditary sensory autonomic neuropathy (HSAN) [28]. Neuropathy can also occur as a feature in diverse genetic disorders with primary involvement of the nervous system, or in metabolic disorders. 

The most common presentation of genetic neuropathy is HMSN. This is a large group of genetic conditions causing progressive dysfunction of motor and sensory nerves. In general, nerves with the longest axons are affected first, producing a length-dependent clinical progression with earlier involvement of the feet and hands. The clinical presentation features symptoms and signs of motor dysfunction including atrophy and weakness of intrinsic muscles in the feet, lower legs, and hands. The longstanding nature of the disorder results in trophic changes in the feet, with distinctive clinical features including pes cavus (high arch) and hammer toes. Although sensory symptoms may be present, typically the motor features predominate in these conditions. Most cases of HMSN are disorders with the sole involvement of the peripheral nervous system; however, other areas of the nervous system (e.g., optic nerves, auditory system, or pyramidal tracts) may be affected [29,30], and the involvement of other organ systems may also occur [31,32]. Furthermore, the neurological deficits from HMSN also result in the secondary involvement of other systems, most commonly the skeletal system (for example, scoliosis [33] and orthopaedic injuries to the feet and ankles [34]) and respiratory system (due to phrenic nerve involvement [35], vocal cord paresis [36], or above-mentioned scoliosis). 

The clinical classification of HMSN/CMT is an evolving area. Classification systems have been based upon clinical presentation, neurophysiologic findings, mode of inheritance, genetic etiology, molecular mechanism, and combinations of these [37]. A detailed discussion of classification systems is beyond the scope of this review, although in practical use it is common for neurologists to describe HMSN based on neurophysiologic features: demyelinating subtype, with reduced conduction velocity, or axonal subtype, with reduced amplitudes and motor-predominant findings. Cases may also be considered as intermediate when features of both demyelinating and axonal pathology are present [38]. This basic classification can be informative to correlate with clinical findings and is also useful in predicting the genetic differential diagnosis. HMSN has a prevalence of 1 in 2500, with more than 80 genes recognized to cause peripheral neuropathy [32,39], although approximately 90% of all cases are attributable to lesions in only four genes (*PMP22*, *MFN2*, *MPZ* and *GJB1*) [40] and the majority of diseases have autosomal dominant inheritance. 

As mentioned above, neuropathy is also a feature of numerous other genetic conditions, and although a detailed discussion is beyond the scope of this work, the subject has recently been reviewed comprehensively [41]. One group of disorders relevant to the content of this review is the presence of neuropathy in numerous metabolic conditions, including mitochondrial disorders [42]. Neuropathy is a central feature of certain mitochondrial syndromes such as ataxia-neuropathy spectrum disorders [43], although neuropathy also occurs as a common feature in milder mitochondrial syndromes [44] and is an under-reported phenotype in patients with mitochondrial disease in general [45]. 

### 1.4. Peripheral Neuropathy, Mitochondrial Dysfunction and Mitochondrial Dynamics

In addition to genetic neuropathy, which is the focus of this review, mitochondrial dysfunction is clearly implicated in acquired neuropathy [46]. For example, oxidative stress resulting from physiological conditions such as diabetes [47], or from exogenous sources such as anti-cancer drugs that damage mitochondrial DNA, is associated with peripheral neuropathy [48,49,50]. However, the exact molecular mechanism underlying neuropathy, be it genetic or acquired, remains undetermined, in part because mitochondria have many cellular functions in addition to their role in generating ATP, and as many types of cellular dysfunction are linked to peripheral neuropathy. Nonetheless, a key aspect of mitochondrial function that has been implicated in causing peripheral neuropathy is mitochondrial dynamics [51]. 

The prevailing theory explaining why impairments in mitochondrial dynamics lead to a peripheral neuropathy phenotype is via reduced mitochondrial motility. The reason being that mitochondrial transport is required for the proper distribution of mitochondria (Figure 1), which is especially important in neurons, as synapses require a continuous and high supply of ATP [15,52,53]. Moreover, the fact that peripheral nerves are very long is thought to make them especially susceptible to reductions in mitochondrial transport. Supporting this idea, the inhibition of both fission and fusion is known to reduce mitochondrial transport [15]. While fission is important to generate mitochondria that are small enough to transport easily, it is less clear why reductions in fusion impair transport. However, recent work suggests the existence of a mitochondrial transport checkpoint that evaluates mitochondrial function prior to allowing axonal transport [54]. As such, dysfunctional mitochondria, which may result from reduced fusion, may be prevented from anterograde axonal transport.

In addition to mitochondrial transport, mitochondrial dynamics impact a number of other cellular functions that are also relevant for peripheral neuropathy. For example, impaired interactions between mitochondria and the endoplasmic reticulum (ER), which regulate calcium signaling and lipid metabolism, have also been linked to peripheral neuropathy [55,56,57,58,59]. Similarly, mitochondria can regulate the formation and utilization of lipid droplets [60], the impairment of which is associated with peripheral neuropathy [61,62,63]. Here, we will review the dynamic processes of mitochondrial fission, fusion and movement, focusing on specific pathogenic variants in key proteins mediating these functions that are linked to peripheral neuropathy. In addition, we will highlight how these proteins impact other aspects of mitochondrial dynamics that are relevant to peripheral neuropathy. Finally, we will also discuss mitochondrial quality control as an unappreciated aspect of mitochondrial dynamics that is relevant to peripheral neuropathy.

## 2. Pathogenic Variants in Proteins Mediating Mitochondrial Dynamics That Cause Peripheral Neuropathy

### 2.1. Mitochondrial Fusion

Mitochondrial fusion is a multistep process that begins with the fusion of the outer mitochondrial membrane (OMM), and which is often followed by the fusion of the inner mitochondrial membrane (IMM) (Figure 2), though these events can be separated functionally. The fusion of the OMM is performed by the mitofusin proteins, MFN1 and MFN2, two homologous dynamin related GTPases that are integral to the OMM [64]. Homodimeric or heterodimeric interactions between MFN1 and MFN2 can tether adjacent mitochondria and mediate their fusion [65]. Although pathogenic variants in MFN2 are one of the major genetic causes of neuropathy, it is somewhat surprising that none have been described in MFN1. The current reasoning for this discrepancy is that the loss of MFN1 function can be compensated somewhat by MFN2. However, as MFN1 is not highly expressed in the nervous system, it cannot rescue the loss of MFN2 functions, which also partially explains why MFN2 dysfunction causes neuronal pathologies. However, as discussed later, we will see that MFN2 also has additional cellular roles, the impairment of which may also explain why pathogenic variants are found in MFN2, but not MFN1.

Meanwhile, IMM fusion is mediated by another dynamin-like GTPase, optic atrophy 1 (OPA1), which is integral to the IMM [66]. In contrast to OMM fusion where mitofusins must be present on both opposing membranes, OPA1 can mediate fusion in vitro when present on a single membrane, though the mitochondrial specific phospholipid cardiolipin must to be present on the opposing membrane [67,68,69]. In addition to these core fusion proteins, a number of other protein factors (many of which have other cellular roles) have been implicated in mediating mitochondrial fusion, including cytosolic factors such as BAX [65] and MSTO1 [21,22], regulators of mitochondrial lipids such as MitoPLD [70], MIGA1 and MIGA2 [71], as well as various other proteins such as FBXL4 [23], ELMOD2 [72], and ARL2 [73]. Here, we focus specifically on MFN2 and OPA1, as pathogenic variants in these genes are clearly related to a peripheral neuropathy phenotype. Additionally, we will examine SLC25A46, which has been described as a negative regulator of mitochondrial fusion [74,75]. Though not discussed in detail here, it should also be noted that motor neuron damage was also recently reported in a single patient with pathogenic variants in MSTO1 [76].

#### 2.1.1. MFN2

Mitofusins contain a GTPase domain, and two heptad repeat domains that are separated by a transmembrane domain. While it was long thought that the N- and C- termini of MFNs were both exposed to the cytosol, recent work suggests that the C-terminus is in fact exposed to the inner membrane space (Mattie et al., 2018). This new understanding of the topology of MFNs has important implications for several of the proposed MFN2 3D structures that have been solved or modelled to date [77,78,79,80]. Critically, these structures are from artificial MFN constructs that lack the transmembrane domain, allowing HR1 and HR2 domains to interact, which would not be the case in vivo. As such, the structural insight from these models is somewhat limited.

MFN2 is a multifunctional protein with several additional roles apart from mitochondrial fusion, including mediating interactions between mitochondria and other organelles (e.g., endoplasmic reticulum (ER), lipid droplets), mitochondrial transport, mitophagy, and even a direct role in lipid transfer to mitochondria. Despite being one of the first additional MFN2 functions to be described, there are still questions about how MFN2 mediates interactions between mitochondria and the ER. While the loss of MFN2 was initially shown to reduce Mt-ER contacts [81], recent observations suggest an increase in Mt-ER contacts [82,83]. While the debate on the role of MFN2 continues [84,85], and it is not clear why this discrepancy exists, the end result is that MFN2 clearly plays a role in mediating Mt-ER contacts. Notably, these contacts between mitochondria and ER (De Brito and Scorrano, 2008) are important in maintaining Ca2+ homeostasis and the exchange of lipids such as phosphatidylserine and cholesterol (Csordás et al., 2018; Krols et al., 2016). More recently, MFN2 was also shown to be involved in mediating interactions between mitochondria and lipid droplets via direct interaction with perilipin, a lipid droplet surface protein (Boutant et al., 2017). The functional relevance of this contact is shown by a recent report showing that the loss of MFN2 in adipocytes leads to an obese phenotype in mice (Mancini et al., 2019). MFN2 is also necessary and directly involved in mitochondrial transport, as it interacts with the Miro/Milton complex that links mitochondria to molecular motors, and the deletion of MFN2 leads to impairment in mitochondrial transport [86,87]. Finally, MFN2 plays a role in mediating mitophagy, as it is directly ubiquitinated by PARKIN [88,89,90] and the loss of MFN2 prevents PARKIN-mediated mitophagy [91]. 

MFN2 was one of the first mitochondrial dynamics proteins linked to peripheral neuropathy [92], with pathogenic variants being the leading cause of autosomally inherited axonal CMT. Over 100 mutations in MFN2 have been reported to be associated with peripheral neuropathy [93], including examples of both autosomal dominant and recessive inheritance (including semi-dominant inheritance). MFN2-related disease is highly variable in its severity, ranging from severe, early-onset neuropathy [94] to very late-onset presentations with low levels of disability [95]. Furthermore, MFN2 mutations can be associated with complex phenotypes in addition to peripheral neuropathy, which occur more commonly with recessive mutations. These phenotypes can include optic atrophy [96,97], hearing loss [96], central nervous system involvement [98], lipomatosis [99,100,101], and ataxia [102]. Given the well-defined role of MFN2 in mediating mitochondrial fusion, it is often assumed that impaired fusion is the primary cause of peripheral neuropathy. However, some MFN2 variants do not have any reported effect on mitochondrial morphology [57,103,104], while other variants actually enhance fusion [105,106]. Thus, the mechanism linking MFN2 dysfunction to peripheral neuropathy is likely not solely due to reduced fusion. In this regard, pathogenic variants in MFN2 have also been shown to impair a number of the other cellular functions mediated by MFN2, including: Mt-ER contacts [57,58,81], mtDNA copy number [97,107,108], lipid metabolism [100], lipid droplets [57], mitochondrial respiration [109], and mitochondrial transport [110]. Notably, altered mitochondrial distribution, including reduced transport and clumping in the cell body, has been described for a number of MFN2 variants (V69F, L78P, R94Q, P251A, R280H and W740S) when expressed in culture rat dorsal root ganglion neurons [110], and is considered to be a major contributing factor to CMT2A pathology [111]. Nonetheless, it is difficult to causally link specific MFN2 dysfunctions with patient phenotypes, as only a few pathogenic variants have been studied functionally, and none have been studied systematically for all known MFN2 functions. Given the multifaceted roles of MFN2 and above-mentioned broad clinical variability, it is also possible that a combination of functions contributes to pathology. Excitingly, small molecules designed to activate MFN2, which restore mitochondrial morphology and motility, have recently been shown to overcome CMT2A mutants in reprogrammed patient motor neurons and in a mouse model [112,113]. While it remains to be determined how this molecule affects other MFN2 functions, it certainly offers a promising therapeutic option.

#### 2.1.2. Optic Atrophy 1 (OPA1)

Like many mediators of mitochondrial morphology, OPA1 also contains a dynamin-like GTPase domain. Alternative splicing can generate multiple isoforms of OPA1, which can be processed into long and short isoforms (l-OPA1 and s-OPA1) via proteolytic processing inside of mitochondria. [114,115]. The processing of l-OPA1 to short forms is induced by a reduction in mitochondrial membrane potential, as well as apoptotic signaling or stresses that induce fission [116]. While l-OPA1 isoforms are essential for fusion, the exact role of the s-OPA1 isoforms remains debated, as they have been implicated in mediating both fusion and fission of the IMM [117,118]. In addition to its role in mitochondrial dynamics, OPA1 has been implicated in binding the mtDNA [119,120] and is also important for maintaining cristae junctions [18,118]. Recent crystal structures of OPA1 have begun to provide a mechanistic basis for how OPA1 can potentially mediate both fusion and fission of the IMM [121,122].

Given the critical role of OPA1 in mitochondrial dynamics and maintenance, it is unsurprising that the deleterious genetic variations present pathologic phenotypes. The most well-studied pathology associated with OPA1 is autosomal dominant optic atrophy (ADOA), where heterozygous pathogenic variants in OPA1 account for over >60% of patients with this phenotype [123,124], and can also cause phenotypes such as optic neuropathy [124], auditory neuropathy [125,126], and peripheral neuropathy [125,127]. Based on the largest described series of OPA1-related ADOA [128], peripheral neuropathy is present in approximately 30% of patients, and is axonal with predominant sensory involvement. The neuropathy commonly co-exists with myopathy and/or ataxia. In general, neuromuscular involvement with ADOA has much later onset (3rd decade) than the ocular features (usually 1st decade), for reasons that are not understood.

In addition to dominant pathogenic variants that cause ADOA, compound heterozygous OPA1 variants are associated with more severe phenotypes. These phenotypes often include peripheral neuropathy, and can present with severe, early, multisystem involvement, such as Behr syndrome, mitochondrial depletion syndrome, or Leigh-like syndrome [129,130,131,132,133,134,135,136]. As a comprehensive understanding of exactly how OPA1 dysfunction causes these varying phenotypes is lacking, the mechanism responsible for peripheral neuropathy specifically remains uncertain. However, it is notable that OPA1 has been implicated in the proper distribution of mitochondria in neurons [54,137,138]. Meanwhile, pathogenic variants in OPA1 lead to the dysregulation of mitophagy [54,131,139,140], with increased mitophagy reported in some dominant negative [131] and biallelic [140] OPA1 variants. Meanwhile, haploinsufficiency variants, including some linked to a peripheral neuropathy phenotype, showed reduced mitophagy [131,139]. These observations suggest that impaired movement and reduced mitophagy may be relevant to the peripheral neuropathy phenotype.

#### 2.1.3. SLC25A46

SLC25A46 is a member of solute carrier family 25 (SLC25) of proteins that consist of three tandem homologous repeats of about 100 amino acids, with each repeat containing two hydrophobic transmembrane regions [141]. While the members of the SLC25 family of proteins are typically integral membrane proteins that transport molecules across the IMM [142], SLC25A46 is present on the OMM and is not believed to have transport activity [143]. Instead, SLC25A46 interacts with proteins involved in mitochondrial dynamics, most notably MFN2 and OPA1 [74,75]. Unexpectedly, SLC25A46 was recognized to be homologous to the yeast protein UGO1, which promotes mitochondrial fusion [143]. However, while SLC25A46 also impacts mitochondrial shape, its loss or impairment stabilizes MFN1/2 leading to hyperfused mitochondrial networks [75]. As such, SLC25A46 is considered to be a negative regulator of mitochondrial fusion. In addition, SLC25A26 has been implicated in mediating mitochondrial cristae dynamics and lipid homeostasis [74].

Pathogenic variants of SLC25A46 are associated with several pathologies previously linked to impaired mitochondrial dynamics. Initially described in patients with an optic atrophy spectrum disorder (including axonal sensorimotor neuropathy) [143,144], SLC25A46 variants are also linked to Leigh syndrome [74], progressive myoclonic ataxia with neuropathy [145], cerebellar ataxia [146], pontocerebellar hypoplasia [147,148] and Parkinson Disease [149]. Several animal models, including mouse [150,151,152,153], zebrafish [143], and *Drosophila* [154], also have neurologic phenotypes that recapitulate human disease. In addition to altered cristae structure and impaired bioenergetics, these animal models show mitochondrial impairment with enlarged mitochondria that have abnormal distribution and which colocalize with the autophagy marker LC3B, consistent with impaired mitophagy. Notably, SLC25A46 knockout mice have peripheral neuropathy with axonal degeneration and demyelination [152]. Pathogenic variants in SLC25A46 can be divided into two classes, those that destabilize the protein and those that alter molecular interactions [155]. Variants that result in the lowest levels of protein expression correlate with the most severe patient phenotypes. The peripheral neuropathy phenotype, which is present in most patients with pathogenic SLC25A46 variants, is thought to be due to a combination of increased fusion leading to hyperfused mitochondrial networks and altered mitochondrial distribution.

### 2.2. Mitochondrial Fission

Fragmentation of the mitochondrial network is critical for the generation of smaller mitochondrial fragments that can be separated from the larger mitochondrial network (Figure 3). As mitophagy cannot degrade larger mitochondria [156,157], fission is essential for mitochondrial quality control, as it facilitates the segregation for the removal of damaged mitochondria. Similarly, fission is required for the generation of smaller mitochondria that can then be transported throughout the cell. For example, the mitochondrial network fragments during the cell cycle, facilitating the distribution of mitochondria into daughter cells [158]. The primary mediator of mitochondrial fission is DRP1, a cytosolic dynamin-like GTPase that forms an oligomeric ring around mitochondria that constricts to mediate mitochondrial division [159,160]. Notably, while examples of DRP1-independent fission have also been reported [161,162,163,164,165,166], there is little known about how this process is mediated.

There are also several other critical factors that mediate mitochondrial fission. The first step in mitochondrial fission is a pre-constriction of mitochondria that is mediated by the ER wrapping around mitochondrial tubules [167] in order to constrict tubules sufficiently for DRP1 oligomers to assemble into rings [168]. Importantly, the force for this initial ER-mediated constriction is generated by the actin-myosin cytoskeleton. Two key regulators of this constriction are the actin-nucleating protein Spire1C present on mitochondrial membrane [169], and the actin polymerizing protein inverted formin2 (INF2) present on ER membrane [170]. Together, these two proteins mediate the formation of actin filaments around mitochondria [171], which then interact with Non-Muscle Myosin II (NMII) proteins to exert a force required for constriction [167,168]. Notably, all three NMII homologs (NMIIA, B, and C) have been implicated in mediating mitochondrial fission [172,173].

Next, there are several distinct proteins anchored to the OMM that help mediate DRP1 recruitment and assembly for mitochondrial fission [174,175,176]. The first DRP1 adaptor to be identified was Fis1 (Fission factor1) [177,178]. Although Fis1 is essential for fission in yeast, it appears to be dispensable in mammalian cells [179,180]. Nonetheless, Fis1 has been implicated is fission events linked to certain stresses [181], as well as also mediating mitophagy [182,183]. Meanwhile, MFF (mitochondrial fission factor) as well as MID49 and MID51 (mitochondrial division proteins of 49 and 51kDa, respectively) seem to play more prominent roles in mediating fission, as their loss leads to hyperfused mitochondrial networks [179,184,185]. Finally, GDAP1 (Ganglioside-induced Differentiation-Associated Protein 1) has also been implicated in mediating mitochondrial fission [186,187], possibly via DRP1 recruitment. Importantly, there is still much to be learned about these distinct adaptors and their relative roles in mediating mitochondrial fission. In addition to mediating mitochondrial fission in response to different physiological signals, another possibility is that different adaptors have tissue-specific roles. In this regard, it may be notable that a pathogenic variant MID49 is associated with an isolated mitochondrial myopathy, without any neuropathy phenotypes [25]. Meanwhile, two dominant pathogenic variants in MID51, which cause impaired fusion/fission dynamics, were recently linked to optic neuropathy [188].

The final step in mitochondrial fission is the scission of the mitochondrial membranes. Structural models suggest that DRP1 can only constrict mitochondrial tubules down to about 50nm, and it is unclear whether this distance is sufficient to completely sever mitochondria or whether other factors are involved. One protein implicated in this final step, DNM2, was initially proposed to be essential for completing fission [189] as it colocalizes on the OMM with DRP1 puncta, and as its knockdown leads to hyperfused mitochondrial networks with constricted regions enriched in DRP1. However, two subsequent studies showed that mitochondrial fission can still occur in knockout cells lacking DNM2, as well as DNM1 and DNM3 [190,191]. While these studies suggest that DRP1 is sufficient for completing mitochondrial fission, the reduced fission in the DNM2 knockout cells [191] and knockdown cells [189] suggests a role for DNM2 in mediating mitochondrial fission, even if it is not essential. 

#### 2.2.1. Ganglioside Induced Differentiation Associated Protein 1 (GDAP1)

GDAP1 is anchored in the OMM via a c-terminal trans-membrane domain [186,192]. While GDAP1 harbors two glutathione S-transferase (GST) domains, initial reports suggested that GDAP1 does not possess GST activity [193,194]. However, more recent work showed that GDAP1 does have active GST activity [195], which is implicated in redox regulation [195,196,197]. Notably, GDAP1 forms homodimers and induces membrane curvature [195]. Though first recognized for its role in promoting mitochondrial fission [186], GDAP1 also mediates peroxisomal fission [198]. In addition, GDAP1 regulates mitochondrial fusion [187], is implicated in the regulation of Mt-ER interactions [199,200], and also impacts mitochondrial distribution [200], potentially via interactions with the cytoskeleton [201] or trafficking molecules such as RAB6B and caytaxin [199].

GDAP1 was one of the first regulators of mitochondrial dynamics linked to peripheral neuropathy, and mutations in this gene are the most common cause of autosomal recessive axonal CMT (CMT4A) [40], although intermediate or demyelinating neurophysiologic findings are also well-described [202,203]. Autosomal dominant inheritance is seen in cases with milder, axonal pathology and incomplete penetrance [204]. The disease onset is typically in childhood and commonly has distinctive phenotypic features including vocal cord paresis and proximal weakness. GDAP1-related diseases are more common in Spanish and Finnish populations due to founder effects [205,206].

Given that GDAP1 is expressed in both Schwann cells and neurons, it is likely that the impairment of both cell types can contribute to the pathology. Intriguingly, different pathogenic GDAP1 variants appear to cause peripheral neuropathy via different mechanisms [187]. Recessive variants, most of which are truncations lacking the C-terminal domain that anchors GDAP1 to mitochondria and cause severe early-onset disease, are associated with reduced mitochondrial fission. Meanwhile, dominant variants, which are primarily missense and lead to a milder pathology, cause reduced mitochondrial fusion leading to mitochondrial damage. Notably, pathogenic defects in GDAP1 do not seem to impair peroxisomal fission [198], suggesting peroxisomal impairment is not relevant to the disease phenotype. Further supporting the important role for GDAP1 in neurons, GDAP1 knockout models in mice [196,200,207], fish [208], and *Drosophila* [209] also recapitulate neurological dysfunction. Importantly, mouse knockout models show axonal neuropathy as well as demyelination, and also exhibit larger mitochondria with anomalous axonal distribution [196,200]. Additionally, GDAP1 knockout causes impaired mitochondrial bioenergetics and disturbed calcium signaling [200]. Notably, oxidative stress [196] and impaired mt-ER contacts [200] are also associated with GDAP1 dysfunction. Thus, there are multiple types of cellular dysfunction that likely contribute to peripheral neuropathy in patients with pathogenic variants in GDAP1. 

#### 2.2.2. DRP1

Dynamin related protein 1 (DRP1), encoded by the *DNM1L* gene, is a dynamin-like GTPase, which was initially shown to be involved in mitochondrial fission [210,211,212] and which also mediates peroxisomal fission [213,214]. The DRP1 protein comprises four main structural domains, the N-terminal GTPase domain, the middle domain, the variable domain (also called insert B), and the C-terminal GTPase effector domain (GED). 

Since the first report of a pathogenic variant in *DNM1L* [19], several pathogenic variants in DRP1 have been associated with highly variable neurological phenotypes [215,216,217,218,219,220,221,222,223,224,225,226,227], with some resulting in complex phenotypes and early mortality [19,228,229,230,231]. Meanwhile, a murine model harboring a mutation in *DNM1L* leads to cardiomyopathy [232], a common type of organ dysfunction in complex mitochondrial syndromes [233]. 

Pathogenic variants in DRP1 that cause reduced fission are located throughout the protein and can affect its activity in a variety of ways, including reduced GTPase activity [219] and impaired higher order assembly of DRP1 oligomers [228]. While epilepsy and encephalopathy are common among patients with pathogenic variants in *DNM1L*, there are also reports of peripheral neuropathy in some patients with heterozygous mutations in the GTPase domains [219,221,234]. Notably, patient fibroblasts harboring the D146N DRP1 variant that causes severe axonal neuropathy exhibit hyperfused mitochondria with reduced mitophagy [219]. Moreover, neuronal loss of DRP1 leads to the accumulation of larger mitochondria with abnormal distribution, consisting of the depletion of axonal mitochondria [235,236,237], while *Drosophila* models also show impaired mitochondrial trafficking when expressing pathogenic DRP1 variants [215,220]. Meanwhile, pain insensitivity, a feature consistent with peripheral neuropathy, has also been reported in two distinct patients with the same G362S heterozygous mutation in the middle domain of DRP1 [216,238]. While the G362S variant causes hyperfused mitochondria, it does not affect peroxisomal morphology, suggesting peroxisomal dysfunction may not be relevant to peripheral neuropathy. DRP1 has also been implicated in the pathogenesis of neuropathic pain in animal models [239]. Together, these observations implicate reductions in mitochondrial motility and mitophagy as likely contributing to the peripheral neuropathy phenotype.

#### 2.2.3. Mitochondrial Fission Factor (MFF)

MFF plays an important role in both mitochondrial and peroxisomal fission [240] by mediating DRP1 recruitment. MFF is anchored to the OMM via its c-terminal transmembrane domain, while the N-terminal domain interacts directly with DRP1 [241]. As evidence of its key role in mediating mitochondrial fission, knockout of MFF prevents DRP1 recruitment to mitochondrial foci and leads to elongated mitochondria [179], comparable to that observed in DRP1 knockdown cells [240]. The phosphorylation of MFF by AMP kinase, a sensor of reduced ATP production and mitochondrial dysfunction [242], promotes mitochondrial fission [243], suggesting that MFF may help coordinate mitochondrial fission in response to energetic stress. 

The importance of MFF for human health is highlighted by the fact that several pathogenic variants in MFF have recently been described [24,244,245,246]. The first report of a pathogenic variant in MFF was a truncating mutation in a patient with delayed psychomotor development, spasticity, Leigh-like encephalopathy and optic atrophy [244]. Meanwhile, other patients with truncating mutations in MFF have also been reported with similar phenotypes as well as peripheral neuropathy [24]. Further highlighting the importance of MFF, knockout mice showed neuromuscular defects, altered gait, decreased grasping ability and reduced fertility [247]. Intriguingly, while MFF -/- mice died at ~13 weeks from dilated cardiomyopathy caused heart failure, when crossed with MFN1 -/- mice, which die at embryonic stage, the double MFF -/-; MFN1 -/- mice do considerably better. This rescue elegantly demonstrates the importance of balance fission and fusion, and offers hope that rebalancing mitochondrial fission and fusion may be a viable therapeutic approach.

#### 2.2.4. Dynamin 2 (DNM2)

DNM2 is an essential protein with many cellular roles, including receptor mediated endocytosis, membrane trafficking, and cytoskeleton regulation [248,249,250], which is embryonically lethal when knocked out in mice [251]. Like DRP1, DNM2 is a member of the dynamin family of large GTPases. It consists of a GTPase domain, a middle domain, a pleckstrin homology (PH) domain, a GTPase effector domain (GED), and a C-terminal proline/arginine-rich domain (PRD). While the PH domain is responsible for membrane binding, the GED functions as a GAP for GTPase activity and mediates the higher order assembly of DNM2. 

Despite the fact that DNM2 does not appear to be essential for mitochondrial fission [190,191], this does not mean that it is not involved in the process. Moreover, there could be certain cell types where DNM2 is more important for fission. In this regard, it is notable that pathogenic variants in DNM2 are well known to cause peripheral neuropathy [252,253,254,255,256], as well as centronuclear myopathy (CNM), characterized by weakness and slow progressive muscle loss [253,257,258,259]. Interestingly, DNM2 variants associated with peripheral neuropathy are mostly located in the membrane binding PH domain [252,253,256]. Meanwhile, variants present in other regions are associated with CNM, with occasional instances of co-existing neuropathy [252,256]. Studies of cell-type specific conditional DNM2 knockout mice provide interesting insight, with Schwann cell knockout resulting in cell death and rapid progression to peripheral neuropathy [260], while oligodendrocyte knockout does not cause significant effects in the CNS [260]. Furthermore, muscle cell specific DNM2 knockout mice had both enlarged mitochondria and increased numbers of lipid droplets, with altered neuromuscular junctions leading to peripheral nerve degeneration [61]. Overall, these cellular phenotypes are consistent with the notion that DNM2 variants impair mitochondrial fission, though this still needs to be confirmed experimentally. 

#### 2.2.5. Inverted Formin 2 (INF2)

INF2 is a member of the formin family of proteins that accelerate actin filament nucleation and elongation [261]. Alternative splicing leads to two isoforms of INF2, one of which is cytoplasmic and regulates the Golgi architecture [262], while the other is anchored to the ER via C-terminal anchors [263]. The ER-localized pool of INF2 regulates the actin-mediated ER-constriction that initiates mitochondrial fission, as the loss of INF2 function leads to reduced numbers of mitochondria-ER contacts and less mitochondrial fission upon ionomycin stimulation [264,265]. In addition to the formin homology 1 (FH1) and 2 (FH2) domains that are characteristic of all formins [266], INF2 is regulated by autoinhibition mediated by the interaction between the C-terminal diaphanous autoregulatory domain (DAD) and N-terminal diaphanous inhibitory domain (DID), and is thus a member of “diaphanous formins” [267,268]. Interestingly, INF2 is unique among formins as it has been found to accelerate both the polymerization and depolymerization of actin filaments. 

Genetic variants of INF2 are associated with autosomal dominant intermediate CMT and glomerulosclerosis, with sensorineural hearing loss commonly present [269,270]. The majority of the pathogenic variations in INF2 are localized to a region in the DID domain [269,271], where they are thought to impair the inhibitory interaction with the DAD domain and render INF2 constitutively active [272]. In the context of mitochondrial morphology, the introduction of the A149D functional mutation in the DID region of INF2, which constitutively activates the protein, led to increased rates of mitochondrial fission and reduced mitochondrial length [170]. Importantly, this work showed that sequence changes in this region can indeed impact mitochondrial morphology. However, unlike other mitochondrial fission proteins linked to peripheral neuropathy, the impairment of INF2 may actually increase fission. Additionally, mitochondrial motility was also reduced in cells expressing the A149D mutation, which is also likely relevant to the peripheral neuropathy phenotype. Notably, recent work shows that like the A149D mutation, pathogenic variants in INF2 cause aberrant formin activity [273], suggesting they may have similar impacts on mitochondrial fission. Nonetheless, as INF2 has multiple isoforms and impaired actin function can impact several cellular functions, it is not clear exactly how pathogenic variants in INF2 cause disease. However, given the role of INF2 in regulating mitochondrial fission, and the fact that unbalanced mitochondrial dynamics and reduced motility are linked to peripheral neuropathy, it seems likely that pathogenic variants in INF2 affect mitochondrial fission [170], although this idea remains to be verified experimentally.

#### 2.2.6. Non Muscle Myosin II C (NMIIC)

Non muscle myosin family II proteins are expressed in all cell types, where they are required for a variety of cellular functions mediated by actin-myosin generated force, including cytokinesis, cell migration, cell shape changes, the internalization of cell surface receptors and mitochondrial fission. There are three NMII family members in humans, NMIIA, NMIIB, and NMIIC, encoded by *MYH9*, *MYH10* and *MYH14*, respectively. All three proteins are believed to be able to form homo or heterodimers with partially redundant functions. Importantly, most cell types express at least one or two NMII proteins, with neurons primarily expressing NMIIB and NMIIC [274]. Although NMIIA and NMIIB are the most studied members of the family and were the first members shown to be involved in mitochondrial fission [172], NMIIC was also recently shown to mediate this process [173].

A number of different loss of function pathogenic variants in MYH14 cause non-syndromic autosomal dominant hearing loss [275,276,277,278,279,280]. However, a single pathogenic variant in MYH14, R941L, which was identified in several unrelated families, is linked to axonal peripheral neuropathy, myopathy, and hoarseness in addition to hearing loss [173,275,281,282]. It is most likely that the hearing loss in R941L patients is due to the loss of NMIIC function, as is the case with most other MYH14 variants. However, the R941L variant in NMIIC also has a dominant negative effect that impairs mitochondrial fission [173]. Notably, it remains to be determined whether or not MYH14 variants that are not associated with peripheral neuropathy also impact fission. Given that patient fibroblasts harboring the R941L variant have hyperfused mitochondrial networks, it is most likely that fission impairments are responsible for the peripheral neuropathy phenotype. It is also intriguing that mitochondrial networks at the cell periphery of MYH14 fibroblasts are resistant to fission, suggesting the NMIIC may have a specific role in this subcellular region. While additional validation is still required, one proposed mechanism is that hyperfused mitochondrial networks in the cell periphery could impair retrograde transport and/or mitophagy, contributing to the disease phenotype [283]. 

### 2.3. Mitochondrial Transport

Mitochondria are transported throughout the cell by an active process that is mediated by molecular motors and the cytoskeleton (Figure 4). In neurons, mitochondrial transport along axons is essential for proper function, as healthy mitochondria need to be mobilized to specific sites of high energy demand, while mitochondria also return to the soma [284]. Neuronal mitochondrial transport occurs along microtubules, with several kinesin proteins capable of moving mitochondria in an anterograde direction (towards the cell periphery), while dynein mediates retrograde mitochondrial transport (towards the nucleus from the cell periphery) [285,286]. Kinesin motors attach to mitochondria via adaptor complexes, the best characterized of which comprises either of the Miro1 or Miro2 homologs, which are atypical Rho GTPases present on the outer mitochondrial membrane, and either of the TRAK1 or TRAK2 homologs, which link Miro1/2 to kinesins. Meanwhile, the mechanism by which dynein interacts with mitochondria remains largely unknown, although TRAK2 has been implicated in dynein-mediated transport [287]. Above and beyond a role in mediating mitochondrial fission, the actin-myosin cytoskeleton can also mediate short range mitochondrial movement. For example, the OMM protein Myo19, which interacts with Miro proteins, can facilitate the transport of mitochondria over short distances along actin filaments [288]. Moreover, actin-myosin interactions can also anchor mitochondria in place and oppose dynein and kinesin-mediated transport [289]. 

In addition to the transport machinery described above, it is also worth noting that key mitochondrial fission and fusion proteins are also implicated in mitochondrial transport. For example, the loss of DRP1 impairs mitochondrial transport [235,290]. While this reduced motility is likely due to reduced fission and mitochondria being too large to move efficiently, it is also worth noting that DRP1 has been implicated specifically in dynein-mediated retrograde mitochondrial transport [291]. Meanwhile, MFN2 plays a direct role in motility via interaction with Miro/TRAK complexes, with reduced mitochondrial motility in cells lacking MFN2 [87]. Notably, several pathogenic MFN2 associated with peripheral neuropathy variants (e.g., V69F, L76P, R94Q, P251A, R280H, W740S) have been demonstrated to impair mitochondrial transport [110].

As discussed above, impaired global axonal transport is one of two main dysfunctions underlying peripheral neuropathies. The fact that the specific inhibition of mitochondrial transport seems to be sufficient to cause of peripheral neuropathy highlights the importance of these critical organelles. It is also worth noting that both anterograde and retrograde mitochondrial transport can be impaired in neurodegenerative diseases such as Alzheimer disease and amyotrophic lateral sclerosis (ALS) [292,293,294]. In Alzheimer disease, where mostly the CNS neurons are affected, optic neuropathy is more common [295,296]. Notably, optic neuropathy occurs in other neurodegenerative conditions with mitochondrial transport defects such as Parkinson disease [297] and Huntington disease [298]. Meanwhile, in ALS, which affects the peripheral motor neurons, peripheral neuropathy is common. In both cases, it is thought that impaired mitochondrial transport contributes to the neuropathy in these disorders. Here, we review pathogenic variants in the mitochondrial transport machinery that are associated with peripheral neuropathy.

#### 2.3.1. Kinesin

Kinesins are a family of molecular motors that transport cellular cargo along microtubules. To date, 45 kinesin family (KIF) proteins have been identified in mammals and are categorized into 15 sub-families [299]. Among these KIF proteins, kinsein-1 subfamily members KIF5A, B and C (also known as kinesin heavy chain; KHC) are primarily responsible for mediating anterograde mitochondrial transport [300,301]. While KIF5B is expressed ubiquitously [300,302], KIF5A and KIF5C are predominantly expressed in neuronal cells, with KIF5A enriched in CNS and KIF5C enriched in motor neurons [302]. Meanwhile, kinesin-3 subfamily members KIF1B [303] and KLP-6 [304] are additional kinesins that can mediate anterograde mitochondrial transport. 

In the context of disease, several pathogenic variants in KIF5A are associated with a disease spectrum ranging from upper motor neuron involvement to peripheral neuropathy [305]. The most common disease associated with KIF5A mutations is spastic paraplegia type 10, which may present as uncomplicated HSP (hereditary spastic paraplegia) or less commonly complicated HSP accompanied by various additional neurologic features [306]. When presenting as peripheral neuropathy, KIF5A-related disease is characterized by axonal sensorimotor neuropathy, but can variably include central nervous system involvement [305,307,308,309,310,311,312]. Supporting the role of KIF5A in mitochondrial transport, the expression of pathogenic KIF5A variants in both *Drosophila* [313] and zebrafish [314] models leads to reduced mitochondrial transport. Additionally, a single loss of function variant in KIF1B has been described in a family with peripheral neuropathy (CMT2A1), and heterozygous KIF1B −/+ mice exhibit chronic peripheral neuropathy and impaired synaptic vesicle transport [315]. However, it remains to be determined whether this KIF1B variant impairs mitochondrial transport or mitochondrial functions. Finally, pathogenic variants in the related protein KIF1A can also lead to peripheral neuropathy, among other neurological phenotypes [316,317,318], and are implicated in mitochondrial dysfunction [319,320]. However, a role for KIF1A in mediating mitochondrial transport has not been described, and reduced KIF1A expression does not impact mitochondrial motility [321]. Collectively, the recurrence of pathogenic variants in proteins that regulate mitochondrial transport suggests a common mechanistic underpinning. 

#### 2.3.2. Dynein

Dyneins are molecular motor complexes that mediate movement along microtubules in a retrograde direction. Retrograde mitochondrial transport is mediated by Dynein 1 (also called cytoplasmic dynein) [322], a heterologous complex made up of two dynein heavy chains (DHC), two dynein intermediate chains, two dynein light-intermediate changes, and various numbers of light chains [323,324]. While the DHC comprises an ATPase domain, a motor domain, a microtubule binding domain and a stem region that binds to intermediate and light chains [325,326], the intermediate and light chains form a sub-complex that binds to various cellular cargo [327].

Dynein 1 impairment has been implicated in a spectrum of neurological diseases [328], with pathogenic variants in the DHC subunit DYNC1H1 linked to neuromuscular disorders including spinal muscular atrophy [329] and peripheral neuropathy [330], as well as mental retardation [331,332]. However, no pathogenic variants in other Dynein I components such as light, light intermediate or intermediate chains have been reported to cause peripheral neuropathy [333]. Notably, a pathogenic variant in *DYNC1H1* has been linked to CMT2O, a finding that is supported by mouse models showing that nearby mutations in *DYNC1H1* cause a sensory neuropathy phenotype [334,335]. Although the human pathogenic variant linked to peripheral neuropathy has not been characterized in the context of mitochondrial function or transport, fibroblasts from patients with DYNC1H1 variants causing spinal muscular atrophy exhibit fragmented morphology [336]. Meanwhile, the two mouse *DYNC1H1* variants that have been described lead to mitochondrial fragmentation [336] and reduced mitochondrial retrograde transport [337]. Thus, it is expected that the pathogenic human DYNC1H1 variant causing peripheral neuropathy will also impair mitochondrial transport and morphology, though this remains to be confirmed.

## 3. Mitochondrial Dynamics and Quality Control in Peripheral Neuropathy

While impaired mitochondrial transport is typically thought to explain the peripheral neuropathy phenotype associated with pathogenic variants in genes regulating mitochondrial dynamics, there is likely more to the story. For example, while the length of peripheral axons is proposed to explain their sensitivity to reduced mitochondrial motility, the CNS axons of humans, which do not necessarily show degeneration when motility is impaired, can be longer than peripheral neurons in mice, which do exhibit peripheral neuropathy [338]. Thus, impaired transport down long neurons alone is insufficient to explain why peripheral neurons are sensitive to impaired mitochondrial dynamics.

Given that transport problems alone cannot explain why impaired mitochondrial dynamics cause peripheral neuropathy, we propose that mitochondrial quality control is an underappreciated aspect of mitochondrial dynamics that is also relevant to peripheral neuropathy. For example, mitochondrial fusion promotes content mixing, which can dilute the effects of mitochondrial damage [64,339,340]. Moreover, MFN2 is directly involved in mediating mitophagy [341,342]. Meanwhile, OPA1 has also been implicated in mitophagy [343], and pathogenic OPA1 variants alter mitophagy. Additionally, mitochondrial fission is also important for quality control as it is essential for mitophagy, due to the fact that large hyperfused mitochondrial networks are resistant to mitophagy [156,157]. Finally, mitochondrial transport itself, which depends on fusion and fission, is important for quality control in the context of delivering healthy mitochondria for fusion complementation and in potentially removing damaged mitochondria. However, while retrograde transport was once thought to be important for the degradation of mitochondria, there is now strong evidence that mitophagy occurs within axons [344] and that mitochondria are not necessarily transported to the soma for degradation [12]. Nonetheless, pathogenic variants that reduce mitochondrial fusion, fission and motility are all expected to negatively impact mitochondrial quality control, suggesting that quality control may be a unifying theme.

Intriguingly, pathogenic variants in many of the peripheral neuropathy proteins discussed here lead to hyperfused mitochondrial networks (e.g., SLC25A46, DRP1, MFF, GDAP1, MYH14). However, the example of MYH14 is especially interesting, given the possibility the impairing fission only at the cell extremity may be sufficient to cause peripheral neuropathy [173]. In this case, it would be expected that mitochondria could still be transported along axons in an anterograde direction, allowing fusion and content mixing. However, hyperfused mitochondria at the cell periphery would be resistant to retrograde transport and mitophagy (Figure 1f). Given the questions around the role of retrograde transport for degrading damaged mitochondria, this model would further highlight the role of axonal mitophagy for quality control and its impairment in peripheral neuropathy.

While there are multiple mechanisms mediating mitochondrial mitophagy, undoubtedly the best studied pathway is the PINK1-PARKIN pathway, which has been well characterized as rare pathogenic variants in PINK1 and PARKIN cause Parkinson Disease [345]. In addition to mediating mitophagy, PINK1 and PARKIN are also implicated in the production of mitochondrial derived vesicles (MDVs) that can be delivered to lysosomes [346]. As MDVs are generated independently of mitochondrial fission, they represent another distinct mitochondrial quality control pathway [4,347]. Notably, there is increased incidence of peripheral neuropathy in patients with Parkinson Disease [348,349,350,351], emphasizing the pathology of Parkinson Disease beyond the CNS.

In addition to mitophagy and MDVs, another recently described neuronal mitochondrial quality control mechanism that may be relevant to peripheral neuropathy is the release of axonal mitochondria into the extracellular space. Such mitochondrial expulsion has been described in the mammalian CNS [352], in *C. elegans* neurons [353], and has been observed in the context of stresses that damage mitochondria where it appears to complement mitophagy as a quality control mechanism [354]. Furthermore, in retinal ganglion cell axons, and likely throughout the CNS, these released mitochondria can be taken up by astrocytes where they can then be degraded in a process termed transmitophagy [352]. Unexpectedly, transmitophagy appears to be more prevalent than mitochondrial degradation in the soma, suggesting it is very relevant to axonal mitochondrial quality control. Conversely, astrocytes have also been shown to deliver presumably healthy mitochondria to neurons [355]. Thus, astrocytes clearly play an important role in maintaining neuronal mitochondrial quality control. While astrocytes can interact with myelinated axons [356], whether similar transmitophagy occurs in peripheral cells is unknown. However, given the distinct morphology of peripheral nerves, which are tightly packaged by Schwann cells, it is reasonable to assume that astrocytes would have restricted access to axons, and that transmitophagy could be of limited benefit. As such, peripheral nerves may depend more on local axonal mitophagy and retrograde transport of damaged mitochondria, which may partially explain why these neurons are more sensitive to impaired mitochondrial dynamics.

## 4. Discussion

In this review, we examined pathogenic variants in key proteins that mediate mitochondrial fusion, fission, and transport, and which cause peripheral neuropathy (Table 1). However, there are other examples of pathogenic variants causing peripheral neuropathy that also impact these dynamic mitochondrial processes (Table 2). Two examples worth briefly mentioning as they offer more concrete links to mitochondrial dynamics are RAB7 and ATL3. The small GTPase RAB7 is best characterized as an endosomal protein linked to CMT2B [357]. However, RAB7 has also recently been implicated in mitochondrial fission via mediating mitochondrial-lysosomal contacts [358,359] and actin dynamics [360], as well as impacting mitochondrial motility [359] and mitophagy [183]. Meanwhile, pathogenic variants in ATL3, which belongs to the Atlastin family of proteins mediating ER morphology, also cause sensory neuropathy and increase mt-ER contacts leading to impaired axonal mitochondrial distribution [56]. Finally, it also remains to be determined whether other peripheral neuropathy genes also alter mitochondrial dynamics, and how they might impact mitochondrial function.

Importantly, pathogenic variants in genes regulating mitochondrial dynamics impact a number of cellular functions in addition to mitochondrial motility that may be pertinent to peripheral neuropathy. Although we focus here on mitochondrial quality control, there are a few other examples worth briefly discussing. Mt-ER contacts, which are impacted by variants in several peripheral neuropathy proteins (e.g., MFN2, GDAP1, ATL3), are important for mediating mitochondrial function, but are also important for mitochondria fission, motility, and likely by extension quality control. On the other hand, while several mitochondrial fission proteins linked to peripheral neuropathy are also involved in mediating peroxisomal fission (e.g., DRP1, MFF, GDAP1), the fact that some pathogenic variants in DRP1 and GDAP1 do not appear to inhibit peroxisomal fission suggests that impaired peroxisomal fission is not relevant to peripheral neuropathy. Finally, while altered lipid droplets have been described in parallel with impaired mitochondrial dynamics (e.g., MFN2, DNM2), and lipid droplet impairment is also associated with peripheral neuropathy [61,62,63], there is not a good appreciation of whether lipid droplet alterations are upstream or downstream of mitochondrial dysfunction.

Given the limitations of reduced mitochondrial motility as a universal explanation for why the impairment of mitochondrial dynamics causes peripheral neuropathy, we highlight the likelihood that impaired mitochondrial quality control is a contributing factor. In this regard, it is also worth highlighting that mitochondrial quality control is especially important in terminally differentiated neurons that cannot be replaced. Importantly, the notion that impaired quality control plays a contributing role to peripheral neuropathy also fits with the fact that stresses causing mitochondrial damage can lead to acquired peripheral neuropathy (e.g., diabetes and cancer drugs). Additionally, it is relevant that peripheral neuropathy can be caused by pathogenic variants in mitochondrial protein chaperones such as HSPB8 (HSP22) [365], HSPB1(HSP27) [366], and TID1 [367], and that modulating chaperones can improve sensory fiber recovery in diabetic peripheral neuropathy [368]. These observations suggest that peripheral neuropathy occurs when mitochondrial damage overcomes the mitochondrial quality control mechanisms, either due to increased damage or reduced quality control. If this notion is true, a part of the explanation for why peripheral nerves are more sensitive to impaired mitochondria dysfunction could be that the quality control mechanism in peripheral nerves is not the same as for nerves in the CNS. Thus, we propose that reduced mitochondrial quality control is a consequence of impaired mitochondrial dynamics, and that insufficient mitochondrial quality control in peripheral nerves is a common thread explaining why mitochondrial dysfunction causes peripheral neuropathy.

## 5. Conclusions

In this review, we have discussed the important role that gene defects causing mitochondrial dynamic alterations have in causing peripheral neuropathy. Neuropathy is often the primary clinical feature in these disorders, and within this group of diseases there is broad clinical and neurophysiologic heterogeneity. Mitochondrial dynamic alterations are clearly an important etiologic category for peripheral neuropathy, particularly for monogenic disease but also in common complex disorders such as diabetic neuropathy. We have described what is currently known about the disease mechanisms and relationships between these genes. Although we do not have a full understanding of the precise mechanisms and pathogenesis, this fundamental mitochondrial function will hopefully be targetable in future therapies. 

## Figures and Tables

**Figure 1 biology-10-00268-f001:**
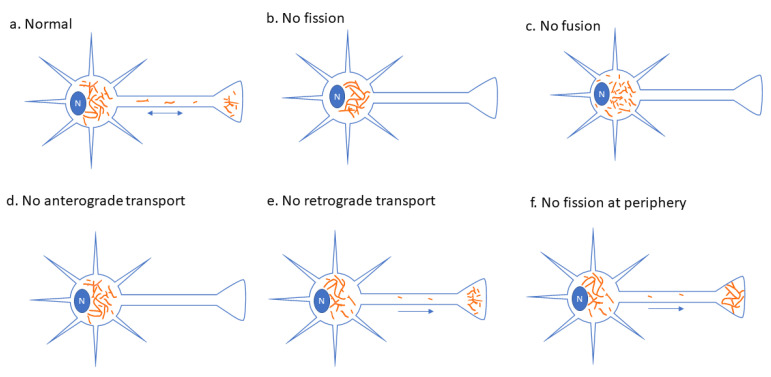
Impact of impairment to mitochondrial dynamics on neuronal mitochondrial transport. (**a**) In a healthy neuron, a variety of mitochondrial morphologies are present and both anterograde and retrograde transport occur. (**b**) In neurons with globally compromised mitochondrial fission, transport is impaired and hyperfused mitochondria remain in the cell soma. (**c**) In neurons with compromised mitochondrial fusion, fragmented mitochondria remain in the cell soma, and their transport is impaired. (**d**) In a neuron with functionally impaired anterograde transport, mitochondria are retained in the soma. (**e**) In neurons with impaired retrograde transport, mitochondria can move down axons and accumulate at the periphery. (**f**) In neurons where mitochondrial fission is impaired in distal regions (e.g., the R941L pathogenic variant of NMIIC), mitochondria are hyperfused at the cell periphery, and retrograde transport is compromised.

**Figure 2 biology-10-00268-f002:**
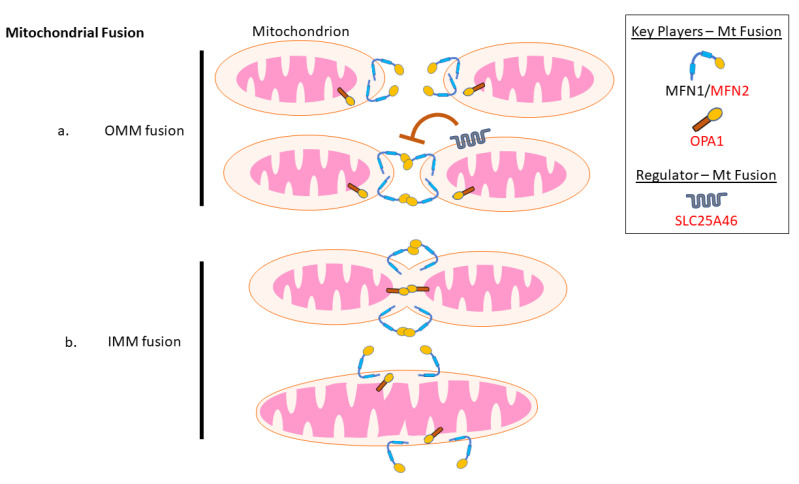
Key mitochondrial fusion players implicated in peripheral neuropathy. Mitochondrial fusion proceeds in two major steps. Proteins highlighted in red are linked to peripheral neuropathy phenotypes (**a**) MFN2, variants in which cause CMT2A, along with its homolog MFN1, mediate tethering and fusion of the outer mitochondrial membrane (OMM). Another OMM protein linked to peripheral neuropathy, SLC25A46, inhibits the actions of MFN1/2. (**b**) The inner mitochondrial membrane (IMM) protein OPA1 carries out IMM fusion. Though best known for autosomal dominant optic atrophy, pathogenic variants in OPA1 can also cause peripheral neuropathy.

**Figure 3 biology-10-00268-f003:**
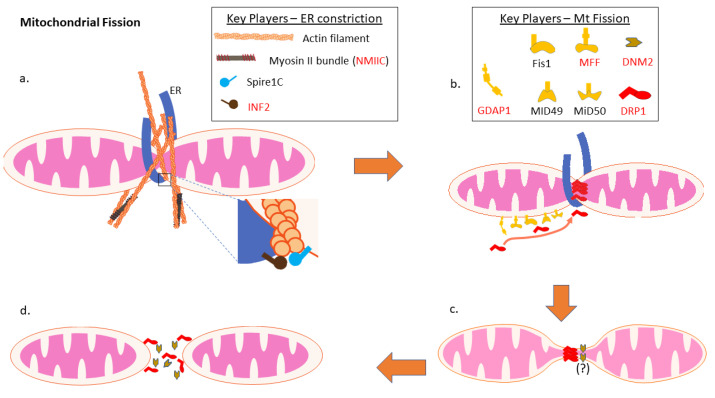
Key mitochondrial fission players implicated in peripheral neuropathy. Mitochondrial fission is a multistep process mediated by several factors. Protein names highlighted in red are linked to peripheral neuropathy phenotypes. (**a**) ER wrapping around mitochondria provides constriction at prospective fission sites that is mediated by actin/myosin motors. Key regulators of this actin-myosin constriction include ER-localized INF2 and mitochondrial-localized Spire1C. Notably, pathogenic variants in INF2 cause peripheral neuropathy and glomerulosclerosis, while a pathogenic variant in MYH14, encoding the NMIIC myosin protein, causes peripheral neuropathy and hearing loss. (**b**) Next, several outer mitochondrial membrane (OMM) proteins (e.g., MFF, MID49/50, and FIS1) act as adaptors that recruit the DRP1 fission protein, which oligomerizes to form a ring around mitochondria. Importantly, pathogenic variants in MFF and DRP1 can cause a variety of neuronal pathologies including peripheral neuropathy. Meanwhile, pathogenic variants in GDAP1, which is also present on the OMM and regulates fission, cause CMT4. (**c**) The oligomeric DRP1 ring tightens to further constrict the mitochondria to set up scission. Though not essential, DNM2 has been also implicated in the final constriction step and is known to harbor pathogenic variants that cause peripheral neuropathy. (**d**) Scission of the constricted mitochondrial tubule leading to the pinching off the two daughter mitochondria.

**Figure 4 biology-10-00268-f004:**
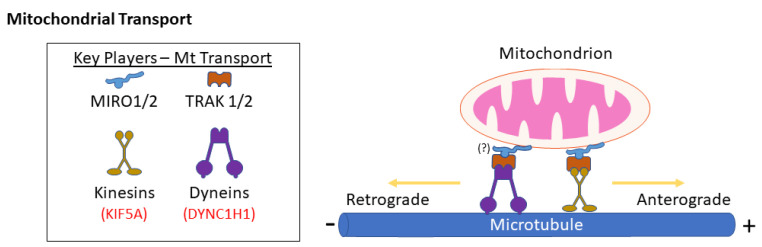
Key regulators of mitochondrial transport implicated in peripheral neuropathy. Anterograde and retrograde mitochondrial transport along the microtubule cytoskeleton is mediated by specific machinery. Protein names highlighted in red are linked to peripheral neuropathy phenotypes. The outer mitochondrial membrane homologs MIRO1 and MIRO2 interact with the homologous TRAK1 and TRAK2 adaptors. Kinesins, including the KIF5A implicated in hereditary spastic paraplegia and peripheral neuropathy, interact with TRAK1/2 to mediate anterograde mitochondrial transport (+). Meanwhile, retrograde mitochondrial transport (−) is mediated by the dynein complex, including DYNC1H1, which is implicated in CMT2O.

**Table 1 biology-10-00268-t001:** Genes encoding proteins that regulate mitochondrial dynamics, and which are implicated in peripheral neuropathy phenotypes either as a primary feature or as an accompanying feature. Clinical classification according to OMIM (Online Medelian Inheritance in Man; www.omim.org, accessed on 1 November 2020). Mt denotes mitochondria, ER denotes endoplasmic reticulum, LD denotes lipid droplets, NMIIC denotes non-muscle myosin IIC.

Gene	Protein Function(s)	Clinical Classification	Clinical Features
Neurogenetic conditions with neuropathy as a primary clinical feature			
*MFN2*	Mitochondrial fusion; Organelle contacts (Mt-ER, Mt-LD); Mitophagy.	CMT2A2A (OMIM 609260) CMT2A2B (OMIM 617087) HMSN6A/CMT6A (OMIM 601152)	Axonal sensorimotor peripheral neuropathy; Optic atrophy; Hearing loss; Lipomatosis.
*SLC25A46*	Negative regulator of mitochondrial fusion.	CMT6B/HMSN6 (OMIM 601152)	Axonal and demyelinating sensorimotor peripheral neuropathy; Optic atrophy; Ataxia; Pontocerebellar hypoplasia; Encephalopathy.
*GDAP1*	Implicated in mitochondrial fission, fusion, motility and ER contacts; Peroxisome fission.	CMT2K (OMIM 607831) CMT2A (OMIM 214400/607706) CMTRIA (OMIM 608340) CMT4A (OMIM 214400)	Axonal, intermediate and demyelinating peripheral neuropathy; Hoarseness (vocal cord paresis).
*DNM2*	Implicated in mitochondrial fission; Vesicle fission.	CMT2B/CMTDIB (606482) LCCS5 (615368) CNM1 (160150)	Peripheral neuropathy; Neuromuscular syndrome; Centronuclear myopathy.
*INF2*	Actin regulator involved in mitochondrial fission.	CMTDIE (OMIM 614455) FSGS5 (OMIM 613237)	Intermediate peripheral neuropathy; Glomerulosclerosis.
*MYH14* (NMIIC)	Mitochondrial fission; Cytokinesis; Cell motility; Cell Polarity.	PNMHH (614369) DFNA4A (600652)	Axonal sensorimotor neuropathy; Myopathy; Hoarseness; Hearing loss.
*KIF5A*	Mitochondrial anterograde transport.	SPG10 (OMIM 604187) NEIMY (OMIM 617235)	Spastic paraplegia; Peripheral neuropathy; Myoclonic seizures.
*DYNC1H1*	Mitochondrial retrograde transport.	CMT2O (OMIM 614228) MRD13 (OMIM 614563) SMALED1 (OMIM 158600)	Peripheral neuropathy; Mental retardation; Spinal muscular atrophy.
Neurogenetic conditions having neuropathy as an accompanying (non-primary) feature			
*OPA1*	Mitochondrial fusion; Cristae organization.	DOA/OPA1 (OMIM 165500) DOA+ (OMIM 125250) Behr Syndrome (OMIM 210000) MTDPS14 (OMIM 616896)	Optic atrophy; Optic neuropathy; Auditory neuropathy; Axonal sensorimotor peripheral neuropathy; Encephalomyopathy; Cardiomyopathy.
*DNM1L* (DRP1)	Mitochondrial fission; Peroxisome fission.	EMPF1 (OMIM 614388) OPA5 (OMIM 610708)	Encephalopathy; Seizures; Peripheral neuropathy.
*MFF*	Mitochondrial fission; Peroxisome fission.	EMPF2 (OMIM 617086)	Encephalopathy; Microcephaly; Seizures; Optic atrophy; Peripheral neuropathy.

**Table 2 biology-10-00268-t002:** Examples of additional genes linked to peripheral neuropathy that also impair mitochondrial dynamics.

Gene	Protein Function	Reported Mitochondrial Dysfunction	References
*RAB7A*	Vesicular transport	Reduced fission, motility and mitophagy	[183,359,360]
*ATL3*	ER network morphology	Increased mt-ER contacts, reduced motility, sparse axonal distribution	[56]
*TRPV4*	Ion channel	Defective mitochondrial motility	[359]
*SIL1*	Co-chaperone in the ER unfolded protein response	Impaired autophagy and mitochondrial maintenance	[361]
*SACS*	Chaperone	Impaired mitochondrial fission	[362,363]
*NEFL*	Axoskeletal component	Aberrant mitochondrial motility	[364]

## Data Availability

Not applicable.

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
