# Peer review of "Genetic Neuropathy Due to Impairments in Mitochondrial Dynamics"

_biology, 2021, doi:10.3390/biology10040268_

Round 1

Reviewer 1 Report

The review by Sharma G. et al. entitled “Genetic Neuropathy due to Impairments in Mitochondrial Dynamics” is a comprehensive description of mitochondrial mechanisms involved in peripheral neuropathy. In particular they focus on the proteins involved in mitochondrial dynamics that can affect either Fission/Fusion process or mitochondrial transport along axons. Overall the manuscript is well written and suitable for publication.

I would add just minor suggestion:

1- Please add YME1L1 as an additional protein involved in mitochondrial fusion (Page 2)

2- Amongst the disease phenotypes caused by OPA1 defect, authors should mention a Leigh-like encephalopathy [Rubegni A et al, 2017] (Page 9)

Author Response

We would like to thank the author for their positive comments and suggestions.

1-With respect to adding YME1L1 as a mediator of fusion, we did not intend to list all proteins with pathogenic variants, just to highlight some of the key proteins that would be discussed in the manuscript. While YME1L1 is a protease the regulates the processing of the fusion protein OPA1, there is still debate about exactly how YME1L1 impacts mitochondrial fusion (or fission), which is beyond the current review. Moreover, YME1L1 has many substrates, and it is not clear that the dysfunction due to pathogenic variants in YME1L1 is specifically due to effects on OPA1 or other YME1L1 substrates. Thus, we have chosen not to include YME1L1 in the manuscript.

2-With respect to OPA1 defects, we have modified the text to include Leigh-like encephalopathy as a feature of OPA1-related disease.

Reviewer 2 Report

Overall, this review is very well done. The authors did an excellent job of summarizing the key aspects of mitochondrial dynamics and peripheral neuropathy and how the two may be linked. In addition to describing some of the well-established mitochondrial dynamics proteins with links to peripheral neuropathy (i.e., MFN2, GDAP1, DRP1) other more underappreciated players in mitochondrial dynamics are also touched on in this review such as NMIIC, INF2, and the motor proteins KIF5A and DYN1H1. The review is organized in a logical way that makes it easy to read and the figures are simple and easy to follow. The secondary focus on the potential role of perturbed mitophagy in peripheral neuropathy also sets this manuscript apart from other similar reviews.

That being said, there are a few things that could be improved:

  • 1 Mitochondrial fusion: Here the authors propose that the reason pathologic variants exist in MFN2 but have never been documented in MFN1 is because MFN1 is not highly expressed in neurons and, therefore, cannot compensate when MFN2 is muted in a neuronal environment. Although this is a reasonable assumption, it is also quite possible that this can be explained by MFN2’s involvement in processes aside from mitochondrial fusion (i.e., mitophagy, mito-ER contacts, mitochondrial transport). Indeed, the authors describe MFN2’s role in these processes later in the review so it’s odd that this possibility it omitted here.

  • 1.1 MFN2: At the end of this section the authors reference a study showing that a small molecule drug that activates MFN2 can rescue the effect of CMT2A mutations in several models. This is indeed exciting and, in fact, is not the only example. Rocha et. al have also demonstrated that an MFN2 agonist is capable of rescuing defects in mitochondrial function and dynamics in the context of CMT2A mutations as well as in a mouse model of CMT.

Rocha AG, Franco A, Krezel AM, Rumsey JM, Alberti JM, Knight WC, Biris N, Zacharioudakis E, Janetka JW, Baloh RH, Kitsis RN, Mochly-Rosen D, Townsend RR, Gavathiotis E, Dorn GW 2nd. MFN2 agonists reverse mitochondrial defects in preclinical models of Charcot-Marie-Tooth disease type 2A. Science. 2018 Apr 20;360(6386):336-341. doi: 10.1126/science.aao1785. PMID: 29674596; PMCID: PMC6109362.

  • Figure 3: The legend states: “Key regulators of this actin-myosin constriction include ER-localized Spire1C and mitochondrial-localized INF2”. It appears the localizations of Spire1C and INF2 have been switched (INF2 localizes to the ER and Spire1C localizes to the mitochondria). In addition, more recent work clarifying the dynamics of actin during mitochondrial fission has been reported in:

Schiavon, C.R., Zhang, T., Zhao, B. et al. Actin chromobody imaging reveals sub-organellar actin dynamics. Nat Methods 17, 917–921 (2020). https://doi.org/10.1038/s41592-020-0926-5

  • 2.1 GDAP1: The authors go into great detail about the role of GDAP1 in mitochondrial fission as well as other aspects related to mitochondrial dynamics such as mito-ER contacts and oxidative stress; however, they do not mention that GDAP1 was also shown to regulate mitochondrial bioenergetics and calcium signaling in a mouse model.

Barneo-Muñoz M, Juárez P, Civera-Tregón A, Yndriago L, Pla-Martin D, Zenker J, Cuevas-Martín C, Estela A, Sánchez-Aragó M, Forteza-Vila J, Cuezva JM, Chrast R, Palau F. Lack of GDAP1 induces neuronal calcium and mitochondrial defects in a knockout mouse model of charcot-marie-tooth neuropathy. PLoS Genet. 2015 Apr 10;11(4):e1005115. doi: 10.1371/journal.pgen.1005115. PMID: 25860513; PMCID: PMC4393229.

  • 2.5 INF2: At the end of this section, the authors state that it is currently unknown how pathologic mutations in INF2 cause disease. Although it is true that mitochondrial dynamics have yet to be studied in the context of pathologic INF2 mutations, recent work has demonstrated that INF2 mutations causing FSGS and CMT cause aberrant formin activity, suggesting that pathologic INF2 mutations may behave similarly to the A149D mutant described in Korobova et. al.

Bayraktar S, Nehrig J, Menis E, Karli K, Janning A, Struk T, Halbritter J, Michgehl U, Krahn MP, Schuberth CE, Pavenstädt H, Wedlich-Söldner R. A Deregulated Stress Response Underlies Distinct INF2-Associated Disease Profiles. J Am Soc Nephrol. 2020 Jun;31(6):1296-1313. doi: 10.1681/ASN.2019111174. PMID: 32444357; PMCID: PMC7269351.

  • Table 2: The first column is slightly too narrow – several of the gene names are split into multiple lines as a result.

  • Table 2: Stating that NEFL increases mitochondrial motility is a bit of an over-simplification. The cited paper does state that NEFL knock-out and NEFL mutations increase mitochondrial motility; however, it also states that the increase in motility is then followed by a drastic decrease such that the mitochondria hardly move at all. They also attribute the initial increase in motility to “wiggling” and not unidirectional transport. “Aberrant mitochondrial motility” may be a more accurate description.

Author Response

We thank the reviewer for their thorough review and excellent suggestions.

1-With respect to the issue of MFN1 vs MFN2 in the context of disease, we certainly agree with the reviewer’s comments. At this point in the paper, we merely wanted to highlight the predominant idea that is found in the literature (e.g. neurons don’t express MFN1). We have modified the text to allude to other alternatives that will be discussed in more detail later.

1.1-As suggested, we have updated the references to include the Rocha paper.

Fig 3-We thank the reviewer for catching this mix-up in the figure legend. We have corrected the figure legend and have added the suggested reference to the main text.

2.1-In the context of GDAP1, we did not initially discuss calcium dynamics, as this is likely a downstream consequence of alterations to mitochondrial dynamics (e.g. morphology and/or mt-ER contacts), rather than a direct effect on calcium. However, we have slightly modified the text to address these issues.

2.5-With respect to INF2, we have updated the text to include the reviewer’s suggestion and new reference.

Table 2-We thank the reviewer for pointing out this formatting error during submission process, which we hope will be corrected in the final manuscript

Table 2-We agree with the reviewer regarding this point on NEFL, and have updated the description of the mitochondrial dysfunction to be more accurate of the initial report.

Reviewer 3 Report

This narrative review explains the role of mitochondrial dynamics in genetics neuropathy. The article is interesting, it addresses a current topic. In my opinion, this work is acceptable.

  • The title and abstract reflect the content of the work.
  • The manuscript is concisely written, and the conclusions drawn are supported by the data and the adequate referencing of past studies.
  • The study is scientifically sound and advances further the knowledge in this very important area of neuropathy.

Author Response

We would like to thank the reviewer for their positive comments on the manuscript.